# Surround Modulation: A Bio-inspired Connectivity Structure for Convolutional Neural Networks

**Hosein Hasani**
Department of Electrical Engineering
Sharif University of Technology
hasani.hosein@ee.sharif.edu

**Mahdieh Soleymani Baghshah**
Department of Computer Engineering
Sharif University of Technology
soleymani@sharif.edu

**Hamid Aghajan**
Department of Electrical Engineering
Sharif University of Technology
aghajan@ee.sharif.edu

## Abstract

Numerous neurophysiological studies have revealed that a large number of the primary visual cortex neurons operate in a regime called surround modulation. Surround modulation has a substantial effect on various perceptual tasks, and it also plays a crucial role in the efficient neural coding of the visual cortex. Inspired by the notion of surround modulation, we designed new excitatory-inhibitory connections between a unit and its surrounding units in the convolutional neural network (CNN) to achieve a more biologically plausible network. Our experiments show that this simple mechanism can considerably improve both the performance and training speed of traditional CNNs in visual tasks. We further explore additional outcomes of the proposed structure. We first evaluate the model under several visual challenges, such as the presence of clutter or change in lighting conditions and show its superior generalization capability in handling these challenging situations. We then study possible changes in the statistics of neural activities such as sparsity and decorrelation and provide further insight into the underlying efficiencies of surround modulation. Experimental results show that importing surround modulation into the convolutional layers ensues various effects analogous to those derived by surround modulation in the visual cortex.

## 1 Introduction

The classical receptive field of a neuron is determined by the region of sensory space where stimuli elicit neural responses. In a sizable population of neurons in the visual cortex, the classical receptive field is surrounded by the nonclassical receptive field, through which the same stimulus can influence the neural response away from that of a mere classical receptive field response [20, 2]. This effect is often referred to as surround or contextual modulation and has been frequently reported in the mammalian visual cortex [20, 2, 29]. The strength of suppression depends on the disparity between the visual features of the stimuli in the classical and nonclassical receptive fields. Various visual features can induce surround suppression, such as spatial frequency, orientation, color, direction of motion, and luminance [2, 29, 59, 53]. Surround modulation has a maximum effect when the center and surround carry similar features. Thus, neurons become more sensitive to the contrast and less sensitive to constant features, and this may enhance the visual perception of objects.

The neural mechanism behind surround modulation is a matter of debate in the literature. However, in general, three types of connections have been identified to be involved in the modulation of

classical and nonclassical receptive fields [3]. Bottom-up feedforward connections from the lower areas affect the center of the receptive field, while intra-areal lateral connections mainly contribute to near surround inhibition, and inter-areal top-down feedback connections from higher cortical areas modulate wider areas including far surround.

Surround modulation is believed to be involved in numerous functions of mammalian visual systems. It plays a fundamental role in boundary detection, contour integration, perceptual grouping, figure-ground segmentation, and border ownership [46, 12, 15, 31, 50]. It is also incorporated in visual attention [25, 56], contrast gain control [18], and perception of depth and motion [2, 6, 5, 26]. In addition to these functions, surround modulation also has notable effects on the neural coding of the visual cortex. It increases sparsity, trial-to-trial reliability and temporal precision of spikes, and decreases temporal and spatial redundancies present in natural scenes by removing predictable components [48, 57, 58, 17].

Theoretical studies have shown that natural scenes can be represented by sparse codes [43, 44]. During natural vision, surround modulation forms a sparse representation in the visual cortex [57, 58, 17]. When image patches encompass both classical and nonclassical receptive fields, modulation becomes more suppressive and the mean spiking rate of individual neurons significantly decreases. During normal vision, wide-field stimulation makes neurons operate at optimum efficiency. Sparse coding increases neural selectivity, which means that each cortical neuron is generally silent but strongly responds to a limited set of visual patterns [45, 62]. Surround modulation strongly decorrelates responses across the population of neurons. This suggests that neurons carry more independent information and are unlikely to fire simultaneously [57].

Further electrophysiological studies have revealed that surround modulation, and hence sparse coding, dramatically increase the average information per spike, information transmission rate, and bandwidth efficiency [58]. On the other hand, dense codes are statistically redundant and metabolically inefficient because in this regime, each neuron responds to a broad set of stimuli, and thus the information is distributed across the neural population, rendering each spike less informative. Sparse coding also enhances the formation of synaptic connections, learning rate, and memory capacity [4, 62]. Together, these properties indicate that surround modulation is an essential mechanism for visual processing by offering informationally, computationally, and metabolically efficient neural codes.

There are historical links between the evolution of CNNs and neurophysiological findings. Hubel and Wiesel first described the concept of receptive fields in visual neurons and proposed a hierarchical structure for the visual cortex by introducing simple, complex, and hypercomplex cells [19, 21]. Motivated by these ideas, the first generations of CNNs were reported in the literature of artificial neural networks [13, 32, 33, 49]. CNNs consist of stacked convolutional layers. In each layer, features are extracted from the input to that layer through local convolution operations. Convolutional layers are followed by pooling operations, which make the extracted features invariant to the geometric transformations of the input [33, 49]. Across the layers, receptive fields become more extensive, and features become more abstract. Thus, a deep structure composed of simple convolution layers can achieve invariant object categorization [30]. Analogously, the visual cortex is composed of a hierarchical organization of distinct cortical areas [11]. Neurons in the early visual cortex have small receptive fields and capture low-level visual features. Along the ventral stream, receptive fields become larger, and neurons respond to more complex features. Finally, in the inferior temporal cortex, neurons respond to the semantic contents of the visual stimuli [22]. Recent studies have shown that this resemblance is not limited to the structure of these networks, and there are also strong correspondences between hierarchical features in the layers of CNNs and neuronal responses in the ventral visual stream with respect to the same stimuli. These correspondences exist even though the CNNs were not optimized to fit neural data and were merely optimized for object recognition tasks [7, 27, 61, 16].

In this paper, motivated by the impressive properties of surround modulation in the biological visual system, we introduce a structure of connections for the standard CNNs to achieve architectures even more similar to the brain. We add local lateral connections to the activation maps of convolutional layers which imitate the function of surround modulation. We found that the proposed structure can outperform traditional CNNs in object recognition tasks while being also capable of speeding up the training procedure. Further evaluations on challenging tasks show that the incorporation of lateral connections can increase the generalization of networks on new domains with different visual

situations which were never seen during training. We also analyze the statistics of the feature space and compare the results with the effects of surround modulation in neural coding of the visual cortex.

**Related Work**   Recently, seeking biologically inspired CNN architectures has drawn considerable attention. Most of the reported studies have focused on augmenting the structure of CNNs by adding recurrent and feedback connections to the convolution layers [41, 54, 36]. One possible explanation for the success of these models is that recurrence enlarges the effective receptive field and also spreads operations across time as well as the network structure. This allows these networks to process each piece of data several times, rendering them capable of reaching the performance of feedforward networks even with shallower models [37, 63, 38]. Linsley et al. [38] proposed a model based on gated recurrent units (GRUs) [8] with additional horizontal connections to solve the contour integration task. Spoerer et al. [54] added between-layer and within-layer recurrent connections to the CNN and evaluated the performance of their model under the presence of noise and clutter on an artificial digit dataset. Nayebi et al. [41] extended the idea by adding attributes such as gating and bypassing, long-range feedback links, and more considerations about the timing of neural dynamics. They also performed a thorough search among different architectures to select the most accurate one.

Our proposed method basically differs from the other biologically motivated models by not using recurrent connections with temporal dynamics. Instead, it employs the existing knowledge about the neural mechanism of surround modulation to incorporate a simple, hard-coded linear operation in the CNNs with no additional training parameters or fundamental changes in the structure of feedforward CNNs.

## 2   Surround Modulation in CNN

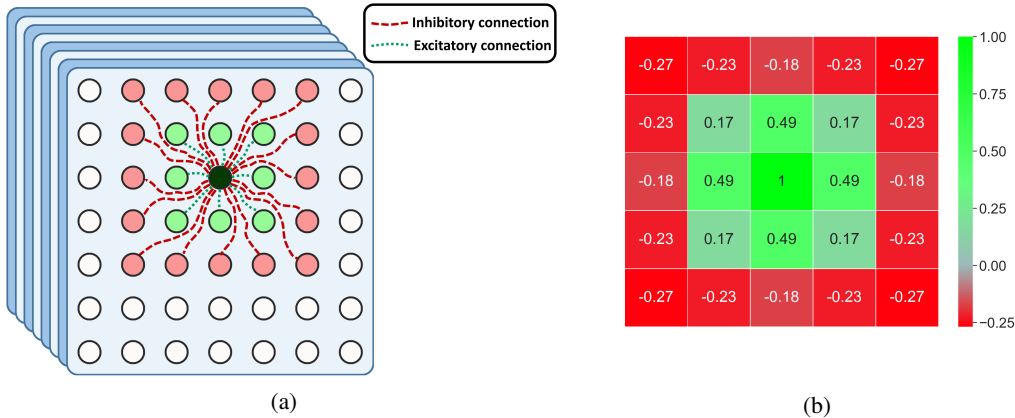

Figure 1: A simple implementation of surround modulation through lateral connections. **(a)** During modulation, each unit excites near neighbors and inhibits far neighbors within a range based on the level of its own activity. **(b)** A $5 \times 5$ SM kernel with associated weights for excitatory connections (green) and inhibitory connections (red).

The Gaussian function has been used to derive a well-established model to describe the mechanism of center-surround modulation [9, 51, 52]. In this model, interactions within the classical receptive field are defined by an excitatory Gaussian, and interactions from the nonclassical receptive filed are defined by a broader inhibitory Gaussian. We employ this model to simulate the center-surround modulation in CNNs through lateral connections. We define a $(2k+1) \times (2k+1)$ kernel whose positive and negative elements respectively determine the excitatory and inhibitory connections between the unit in the location $(k+1, k+1)$ and its neighbors (Figure 1a). The weight associated with each neighbor is defined by the difference of Gaussian (DoG) function:

$$DoG_{\sigma_e,\sigma_i}[x,y] = \frac{1}{2\pi\sigma_e^2}exp(-\frac{(x-(k+1))^2 + (y-(k+1))^2}{2\sigma_e^2})- \\ \frac{1}{2\pi\sigma_i^2}exp(-\frac{(x-(k+1))^2 + (y-(k+1))^2}{2\sigma_i^2}). \tag{1}$$

where $\sigma_e$ and $\sigma_i$ are the standard deviations of the excitatory and inhibitory Gaussians, respectively. The surround modulation (SM) kernel is obtained by normalizing the DoG to the amplitude of its center:

$$SM[x,y] = \frac{DoG[x,y]}{DoG[k+1,k+1]}. \tag{2}$$

The SM kernel has been designed to increase the feature saliency by suppressing redundant and spatially constant responses within activation maps. Each unit in the activation map $m$ of layer $l$ is modulated by its spatial neighborhood with the corresponding weights in the SM kernel. One plausible approach is to linearly add the weighted responses of the surround units to the response of the center unit. To do this, it is sufficient to convolve each activation map by the SM kernel to obtain the surround modulated activation map:

$$act_{SM}^{l,m}[x,y] = (act^{l,m} * SM)[x,y]. \tag{3}$$

It is not necessary to strictly follow the exact DoG profile to achieve excitatory connections in the center and inhibitory connections in the surround. Especially for SM kernels with smaller sizes, providing a proper balance between the amounts of excitation and inhibition may entail subtle modifications of the kernel. For each specific task, finding the optimal design for the SM kernel and its most effective placement in the CNN may need performing hyperparameter search. However, the aim of this paper is not to explore all possible setups, but to present a simple, fixed setup which resembles the surround modulation phenomenon in the cortical nervous system. Hence, for all of our experiments we deploy a fixed SM kernel with a size of $5 \times 5$ ($k = 2$) obtained by setting $\sigma_e = 1.2$ and $\sigma_i = 1.4$ (Figure 1b). In our experiments, the SM kernel was added to activation maps of the first convolutional layer as such modulation is more common in the early visual cortex.

## 3 Experiments

To evaluate the performance of the proposed method, we set up three types of experiments. First, we apply the proposed model to a standard image classification task. Then, we evaluate the robustness of the model in different visual situations. Finally, we analyze the characteristics of neural activities in the presence of surround modulation and compare the results with those reported for the visual cortex.

In all experiments, baseline is a standard CNN which consists of multiple convolution layers with $3 \times 3$ kernels and stride of 1, multiple max-pooling layers with $2 \times 2$ kernels and stride of 2, and three fully-connected layers at the end of the network. The last layer predicts the probability of categories with a softmax activation function and the other fully-connected convolution layers have ReLU non-linearities. All implementations are done in Tensorflow [1], and during the training procedure, Adam optimizer [28] with a learning rate of $10^{-4}$ is used to minimize the cross entropy loss (see Supplementary Materials for more details). We train all of the networks from scratch by initializing trainable weights with Xavier initialization [14], and repeat each experiment 10 times.

### 3.1 Image Classification

We used the ImageNet dataset [10] for object recognition as it contains natural images with an acceptable resolution to test surround modulation. For further analysis, we composed a baseline dataset, hereby called baseline-ImageNet, by randomly choosing 100 categories. From each category, 500 instances were randomly chosen for training, 50 instances for validation, and 100 instances for the test set. All images were cropped around their centers and resized to $160 \times 160$ pixels. The baseline network contains 2.3M trainable parameters. It has seven convolutional layers, five pooling layers, and three fully-connected layers with Dropout [55] in each of them. We did not perform any data augmentation during training or any multi-cropping during the test. This was done to provide a standard and balanced dataset along with a standard CNN architecture, i.e. an impartial setting, to evaluate the exclusive effect of adding SM to the network.

We constructed the SM-CNN by adding the SM kernel to half of the activation maps in the first convolution layer of the baseline network. The SM-CNN has the same number of training parameters as the baseline network and requires less than $1\%$ extra computations for performing surround

modulations. Figure 2 shows the validation loss and accuracy as a function of the number of training steps. Surround modulation increases both the training speed and accuracy of the baseline model ($p < 0.001$).

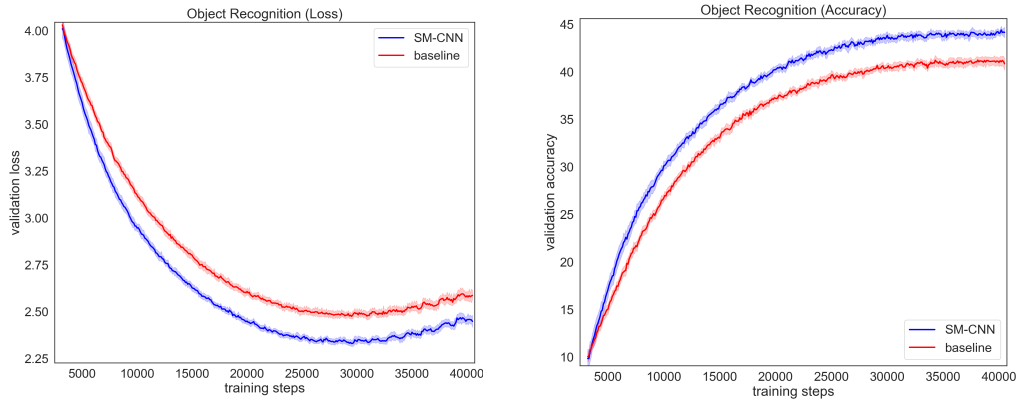

Figure 2: Baseline-ImageNet validation loss (left) and top-1 accuracy (right) of the baseline network and SM-CNN across training steps. The pale margins indicate confidence intervals in the trials.

To compare the performance of the models more precisely, we designed six additional networks as follows. Two control models were designed by adding an extra layer after the first convolutional layer of the baseline network with a kernel size of $(5 \times 5)$. The number of convolution kernels is the same as the number of activation maps in the first layer. The first control model ($\mathbf{C}^1_{\text{baseline}}$) has no activation function after the new layer, while the second one ($\mathbf{C}^2_{\text{baseline}}$) has a ReLU nonlinearity.

Four variants of the SM-CNN were also designed. In the first variant ($\mathbf{C}^1_{\text{randSM}}$), we replaced the SM kernel with a random kernel matching the first and second-order statistics of the SM kernel to analyze the advantages offered by the SM configuration. In the second variant ($\mathbf{C}^2_{\text{SM}}$), the SM kernel was applied to the input image as the preprocessing step. In the third variant ($\mathbf{C}^3_{\text{SM}}$), the SM kernel was added to all activation maps of the first layer. In the fourth variant ($\mathbf{C}^4_{\text{SM}}$), the SM kernel was applied after the first max-pooling layer, which is itself located after the first two layers of convolution. The top-1 accuracy of all the network variants is shown in Table 1. The main SM-CNN, as well as its third and fourth variants, outperform the traditional baseline variants ($p < 0.001$) even with fewer parameters and computations.

Table 1: Top-1 accuracy of the baseline and SM-CNN and their related control models for the baseline-ImageNet dataset.

| | baseline variants | | | SM-CNN variants | | | | |
|---|---|---|---|---|---|---|---|---|
| | main | $\mathbf{C}^1_{\text{baseline}}$ | $\mathbf{C}^2_{\text{baseline}}$ | main | $\mathbf{C}^1_{\text{randSM}}$ | $\mathbf{C}^2_{\text{SM}}$ | $\mathbf{C}^3_{\text{SM}}$ | $\mathbf{C}^4_{\text{SM}}$ |
| ACC | $41.0_{\pm 0.6}$ | $41.3_{\pm 0.6}$ | $40.7_{\pm 0.5}$ | $43.8_{\pm 0.7}$ | $41.0_{\pm 0.9}$ | $40.8_{\pm 0.4}$ | $42.3_{\pm 0.6}$ | $43.2_{\pm 0.4}$ |

The inferior performance of the baseline variants indicates that the surround modulation effect cannot be trivially replaced by a learnable convolution layer. The average performance of $\mathbf{C}^1_{\text{randSM}}$ is not higher than the baseline, indicating the effectiveness of the excitatory-inhibitory pattern in the proposed SM kernel. The performance levels of the third and fourth SM-CNN variants imply that surround modulation could be effective in various setups even when applied to the intermediate layers of the network, while the performance level of the second SM-CNN variant indicates that adding the SM kernel to the input of the network may not provide a notable improvement.

## 3.2 Robustness and Generalization

As discussed in the introductory remarks, surround modulation serves as a proxy function for various visual perception tasks. In this section, we examine the ability of the proposed method in handling challenging visual tasks. We train each model on the standard datasets and test them on domains with different situations as follows.

**Illumination**   Here, we explored the generalization capability of the networks under drastic changes in illumination. To this end, we used the small NORB dataset [35], which contains images of 3D objects under six different lighting conditions (examples in Figure 3a). We trained the networks on training images from one lighting situation and tested them on test sets composed of unfamiliar lighting conditions. To assess whether the generalization capability offered by surround modulation can be obtained by other standard regularization methods, we examined three additional control models. In the first control model, we added $L_2$ regularization to all of the trainable weights of the baseline network. In the second control model, we added Dropout to the last three fully-connected layers of the baseline network. In the third control model, we added Batch Normalization [24] to all convolutional layers of the baseline network.

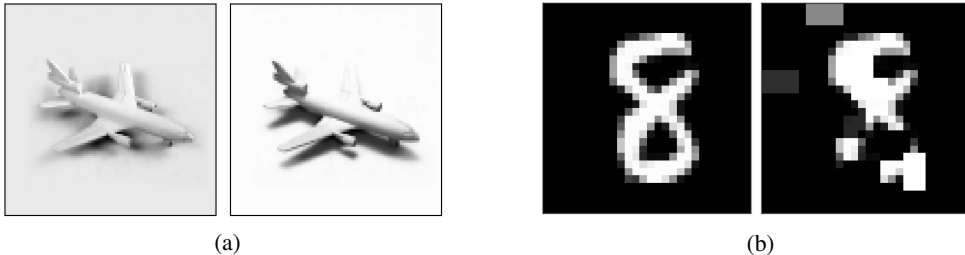

(a)                                                                      (b)

Figure 3: Data samples for challenging image classification tasks. **(a)** An instance from the small NORB dataset with its different illumination variant. **(b)** An instance from the MNIST dataset with its artificially occluded variant.

The overall accuracies on familiar and unfamiliar test sets are reported in Table 2. Averaging across six conditions and ten trials shows that the baseline CNN lost about 36% of its accuracy in new lighting situations, whereas the SM-CNN lost just about 17%. The superior robustness of the SM-CNN over the standard regularization methods suggests that surround modulation can be employed as a regularization technique, although it was not directly designed for this purpose. Figure 4 shows the accuracies of familiar and unfamiliar test cases in one of the lighting conditions as a function of training steps.

Table 2: The overall test accuracy of the familiar and unfamiliar test sets for all six lighting conditions.

|  |  | $\text{light}_0$ | $\text{light}_1$ | $\text{light}_2$ | $\text{light}_3$ | $\text{light}_4$ | $\text{light}_5$ |
|---|---|---|---|---|---|---|---|
| familiar | baseline | $85.2_{\pm 1.6}$ | $85.6_{\pm 1.4}$ | $80.8_{\pm 1.1}$ | $85.6_{\pm 1.2}$ | $85.3_{\pm 1.1}$ | $84.6_{\pm 1.8}$ |
|  | $\text{baseline}_{L_2}$ | $86.8_{\pm 1.5}$ | $86.5_{\pm 1.8}$ | $81.2_{\pm 2.3}$ | $85.8_{\pm 1.2}$ | $85.6_{\pm 1.7}$ | $86.4_{\pm 1.6}$ |
|  | $\text{baseline}_{\text{Dropout}}$ | $88.0_{\pm 1.8}$ | $88.4_{\pm 1.6}$ | $82.7_{\pm 1.8}$ | $84.7_{\pm 2.3}$ | $\mathbf{87.4}_{\pm 1.9}$ | $85.9_{\pm 1.8}$ |
|  | $\text{baseline}_{\text{BN}}$ | $90.8_{\pm 0.8}$ | $91.7_{\pm 1.6}$ | $86.8_{\pm 2.0}$ | $89.0_{\pm 1.6}$ | $85.8_{\pm 0.8}$ | $87.3_{\pm 1.1}$ |
|  | SM-CNN | $\mathbf{92.3}_{\pm 1.1}$ | $\mathbf{93.0}_{\pm 0.8}$ | $\mathbf{89.4}_{\pm 1.1}$ | $\mathbf{91.7}_{\pm 0.9}$ | $87.1_{\pm 1.3}$ | $\mathbf{90.1}_{\pm 1.1}$ |
| unfamiliar | baseline | $57.4_{\pm 1.5}$ | $56.1_{\pm 2.4}$ | $35.1_{\pm 2.1}$ | $55.0_{\pm 1.6}$ | $46.8_{\pm 2.1}$ | $41.3_{\pm 2.0}$ |
|  | $\text{baseline}_{L_2}$ | $59.8_{\pm 1.7}$ | $56.7_{\pm 1.8}$ | $35.5_{\pm 1.4}$ | $56.0_{\pm 2.0}$ | $46.9_{\pm 1.6}$ | $42.5_{\pm 1.5}$ |
|  | $\text{baseline}_{\text{Dropout}}$ | $66.8_{\pm 1.8}$ | $67.4_{\pm 2.4}$ | $36.8_{\pm 1.6}$ | $62.2_{\pm 2.7}$ | $49.1_{\pm 1.1}$ | $48.9_{\pm 3.6}$ |
|  | $\text{baseline}_{\text{BN}}$ | $65.1_{\pm 2.8}$ | $57.2_{\pm 2.7}$ | $31.6_{\pm 3.7}$ | $64.8_{\pm 2.2}$ | $43.1_{\pm 5.1}$ | $50.8_{\pm 1.6}$ |
|  | SM-CNN | $\mathbf{80.8}_{\pm 1.2}$ | $\mathbf{73.0}_{\pm 2.5}$ | $\mathbf{61.4}_{\pm 2.5}$ | $\mathbf{82.2}_{\pm 0.8}$ | $\mathbf{65.8}_{\pm 1.3}$ | $\mathbf{80.3}_{\pm 1.0}$ |

**Occlusion**   A potential advantage of employing surround modulation by the biological visual system is increasing robustness to the presence of occlusion and clutter. Handling these conditions in the brain may be additionally facilitated by attention mechanisms through top-down feedback connections [47]. In our analysis, we investigate the robustness of the proposed surround modulation method in the presence of small clutter on the MNIST dataset [34].

We augmented the test images by adding from 8 to 15 random patches with different shapes and intensities to each image (one example is shown in Figure 3b). The length of each patch varied from 3 to 6 pixels. To estimate the generalization ability of the networks in challenging situations more accurately, we restricted the size of the training set to 1000 by randomly sampling 100 samples from each class. Networks were trained on the standard images with no additional augmentation but tested

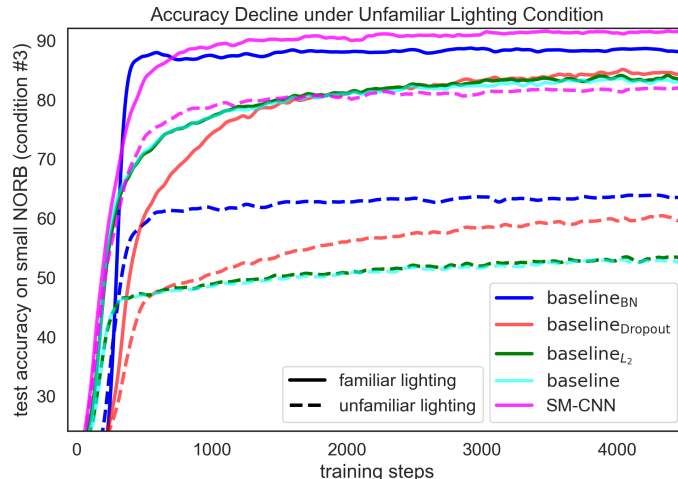

Figure 4: The accuracy of the familiar and unfamiliar test sets with training performed on the lighting condition 3 of the small NORB dataset.

on the standard and occluded images. As shown in Table 3, in the occluded scenario, the accuracy of the baseline method dropped by $37.8\%$ while the accuracy of the SM-CNN dropped by $26.2\%$.

Table 3: Classification accuracy on the standard and occluded test images of the smaller version of the MNIST dataset.

|  | standard | occluded |
|---|---|---|
| baseline | $92.6_{\pm 0.3}$ | $54.8_{\pm 1.1}$ |
| SM-CNN | $93.2_{\pm 0.2}$ | $67.0_{\pm 1.1}$ |

## 3.3 Neural Activity Characteristics

As stated in the introductory section, a particular effect of surround modulation is to increase the sparsity of the neural activities. This effect bodes well with the requirements of biological visual systems which need to operate under maximum information and energy efficiency constraints during natural vision. To assess whether our implementation of surround modulation is capable of inducing sparsity to artificial neural networks, we set up a task-driven approach. We optimized networks for object recognition on natural images and then analyzed the neural activities of the final networks and compared the results with observations reported in studies of the biological visual system. To this end, we deployed the networks which were trained on the baseline-ImageNet dataset and fed them with 5000 images from the test set. Then, we recorded the neural activity of the first and third convolution layers. In addition to the baseline network and the SM-CNN, we also analyzed the effect of the presence of Batch Normalization in the first convolution layer on neural activities. For each model, we trained six different networks from scratch to ensure that the statistics do not change in different trials.

We report sparsity of the neural responses in two different ways: lifetime sparsity, which characterizes the sparsity of the response of each unit to the entire set of stimuli, and population sparsity, which characterizes the sparsity of the activity of the neural population in response to a stimulus [60]. We compute the population and lifetime sparsities with kurtosis [1] and the selectivity index, both of which are well-studied metrics in neuroscience [57, 58, 60, 45]. Figures 5a and 5b show the distributions of the selectivity index for the lifetime sparsity and the kurtosis index for the population sparsity of the third convolutional layer, respectively. We also estimated the sparsity of the overall activities by the Gini metric, which is a suitable sparsity measure especially for one-sided distributions [23] (see

Table 4: Average sparsity scores of neural activities from the first and third convolutional layers across six trials. In all of the sparsity measures, SM-CNN and baseline$_{BN1}$ have higher scores than the baseline network, indicating higher sparsity in their neural coding.

| | First convolutional layer | | | Third convolutional layer | | |
|---|---|---|---|---|---|---|
| | baseline | baseline$_{BN1}$ | SM-CNN | baseline | baseline$_{BN1}$ | SM-CNN |
| kurtosis | $5.8_{\pm 0.7}$ | $22.0_{\pm 1.2}$ | $22.6_{\pm 1.7}$ | $11.6_{\pm 1.1}$ | $17.2_{\pm 2.7}$ | $28.2_{\pm 3.3}$ |
| selectivity | $0.50_{\pm 0.02}$ | $0.77_{\pm 0.01}$ | $0.72_{\pm 0.02}$ | $0.66_{\pm 0.03}$ | $0.71_{\pm 0.07}$ | $0.78_{\pm 0.01}$ |
| Gini | $0.51_{\pm 0.02}$ | $0.74_{\pm 0.01}$ | $0.64_{\pm 0.01}$ | $0.65_{\pm 0.03}$ | $0.67_{\pm 0.09}$ | $0.76_{\pm 0.01}$ |

Supplementary Materials for definitions). In all of the metrics, a higher value indicates higher sparsity. Table 4 shows the sparsity measures of the first and third layers of each model across the population and stimuli. As the results indicate, surround modulation, as well as CNN with Batch Normalization in its first layer (baseline$_{BN1}$), significantly increase the sparsity of the neural activities.

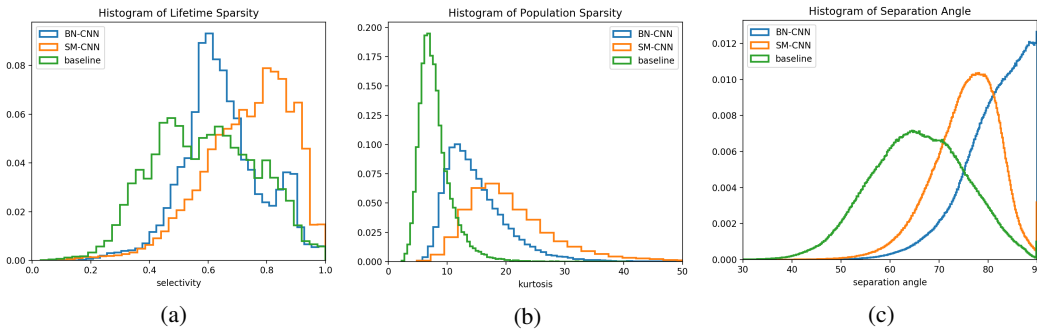

Figure 5: Distributions of sparsity and decorrelation for the third convolutional layer. **(a)** Distribution of lifetime sparsity based on selectivity index. **(b)** Distribution of population sparsity based on kurtosis index. **(c)** Distribution of decorrelation based on separation angles between random pairs of neurons.

To evaluate whether surround modulation increases independence between the units of a specific layer, we followed the same strategy employed in [57]. We randomly selected 1M pairs of neurons in each layer and computed the separation angles between their responses to 5000 stimuli (see Supplementary Materials). Neuron pairs that carry similar information would have low separation angles. Figure 5c shows the distribution of separation angles in the third convolutional layer of the models mentioned above. Surround modulation, as well as Batch Normalization, decrease the correlation between neural activities among the neural population, and this property also propagates to higher layers.

Together, these results indicate that our simple implementation of surround modulation in CNNs offers considerable efficiencies in neural coding of the early and intermediate layers. Surround modulation increases sparseness and decreases statistical redundancies, analogous to similar effects reported in studies of the biological visual system [57, 58, 17]. Our results also provide more insight about efficient neural coding in CNNs and suggest that sparse coding in these networks may offer similar properties as those considered for sensory cortices, such as improvement in the learning rate and prediction accuracy [4, 62].

## 4 Discussion

In the present study, we introduce a biologically plausible modification of CNNs by leveraging a well-known phenomenon in the biological visual system, the so-called surround modulation. The DoG function is a well-studied operator in computer vision, which is mainly used for low-level feature extraction like edge and blob detection [40, 39]. Inspired by the role of this function in modeling surround modulation, we implemented a simplified version of surround modulation by a linear operation that can be easily incorporated into the convolutional layers of CNNs. We found that the proposed method can significantly improve the performance of CNNs in standard image

classification tasks. While the resulting improvement is not trivial, it is also somewhat surprising, because the proposed method merely introduces some linear operations with no additional training parameters, and the increment in computations is also limited.

We investigated the impact of this simple form of surround modulation on improving the robustness of CNNs under difficult visual conditions. We also analyzed the potential implications of the proposed surround modulation structure in increasing sparsity and decorrelation in the neural activities of the CNNs. In most of the cases, the surround modulated CNN reaches the performance of the baseline in fewer optimization steps or with a smaller size of training data. Based on these experiments, it seems that surround modulation is capable of facilitating learning, an observation that is consistent with the neurophysiological studies of the brain [62]. A possible justification is that surround modulation increases feature saliency and decreases redundancies which may slow down training. We also analyzed neural activities in the presence of Batch Normalization which is also considered to improve the accuracy of the network and its training speed. Batch Normalization increases sparsity and decorrelation analogous to surround modulation. Thus, the success behind Batch Normalization presumably can be explained by efficient neural coding.

The potential gain offered by surround modulation on the training speed is interesting as it alleviates the need for numerous training steps on large datasets, which is a major drawback of traditional CNNs in comparison with the biological visual system. However, more meticulous studies are needed to explore the role of surround modulation in facilitating few-shot learning.

Our implementation of surround modulation is straightforward. It can be easily utilized in different CNN architectures and for different visual tasks. Semantic segmentation is one type of challenging tasks in which surround modulation may perform well. Surround modulation reduces sensitivity to constant textures and also gets involved in various functions associated with image segmentation such as border ownership, perceptual grouping, and contour integration [12, 50, 15, 31, 46]. However, in occlusion scenarios, one may need to employ top-down feedback links in order to incorporate more abstract features in the segmentation task [47].

In summary, this work introduces a new biologically-motivated connectivity structure for the CNNs resembling the structure of the visual cortex. As a result, more biologically plausible neural coding, better generalization performance, and more biologically plausible behavior in training speed are also achieved, even though the method was not explicitly designed for these gains. More work is needed to explore and compare various setups for implementing surround modulation. Here, we implemented surround modulation as near-surround lateral connections, but physiological studies have frequently reported the presence of extensive top-down feedback links [42], especially involved in far-surround modulation, which is not covered in the present study and is worth exploring.

## Footnotes

[1]The distribution of the amplitudes of neural responses is one-sided, while kurtosis is suitable for two-sided distributions. So, similar to [57], we add to each distribution its mirrored version around the origin and estimate the kurtosis metric for the resulting two-sided zero-mean distribution.

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
