[Supplementary Material]

# Supplementary Materials: "Surround Modulation: A Bio-inspired Connectivity Structure for Convolutional Neural Networks"

**Hosein Hasani**
Department of Electrical Engineering
Sharif University of Technology
hasani.hosein@ee.sharif.edu

**Mahdieh Soleymani Baghshah**
Department of Computer Engineering
Sharif University of Technology
soleymani@sharif.edu

**Hamid Aghajan**
Department of Electrical Engineering
Sharif University of Technology
aghajan@ee.sharif.edu

## 1   Appendix

### 1.1   Network Architectures

In this section, we provide more details about the configurations used in our experiments.

The size of input images used in the ImageNet, small NORB, and MNIST experiments are 160, 48, and 28, respectively. The dimension of the feature map tensors is shown in Figure 1. For all of the networks, we use convolution kernels of size $(3 \times 3)$, and each pooling layer reduces the spatial dimension of feature maps by a factor of 2. For SM-CNN, surround modulation was applied to all of the feature maps of the first layer in the cases of the small NORB and MNIST tasks, but was applied to half of the feature maps in the ImageNet task. In the case of the ImageNet classification, zero-padding was applied before each convolution layer, and Dropout was performed in each fully-connected layer.

The size of training datasets used in the ImageNet, small NORB, and MNIST experiments are 50,000, 8100, and 1000, respectively. For all of the experiments, we used a batch size of 32, and hyperparameters were roughly tuned for baselines. Our analysis shows that a learning rate of $10^{-4}$ offers a good tradeoff between training speed and robustness for all scenarios. Training cut off was decided upon when further training steps did not yield notable changes in the performance. We used epoch size of 26, 30, and 60 for the ImageNet, small NORB, and MNIST experiments, respectively.

### 1.2   Experiments on Networks with Different Structures

In the paper, we used a standard and general CNN architecture for the baselines to mitigate potential effects associated with complex settings. For example, we observed that the presence of surround modulation along with Batch Normalization, might reduce the gains offered by each one. The gain offered by surround modulation may be affected by factors including details of the network architecture and regularizations, training size, optimization procedure, and the configuration used for surround modulation. For new experiments, the effect of each of these factors needs to be examined. Here, we repeated the ImageNet experiment with a standard ResNet-18 with pre-activation and $L_2$ regularization [1]. We used the same strategy for incorporating the SM kernel, except that we did not apply pre-activation for the activation maps to which the SM kernel was already applied.

Although ResNet extensively uses Batch Normalization, which is expected to reduce the gain offered by surround modulation, our SM-ResNet-18 still showed superior performance over the standard

(a) Network architecture for the ImageNet experiment.

(b) Network architecture for the small NORB experiment.

(c) Network architecture for the MNIST experiment.

Figure 1: Architecture of the baseline network for the ImageNet (top), small NORB (middle), and MNIST (bottom) classification experiments. Pooling layers are shown inside their respective convolutional layers with darker colors.

ResNet-18 (Figure 2a). We repeated this experiment on a smaller training dataset obtained by sampling 100 instances from each class. We observed that SM-ResNet-18 offers about 12% gain in relative accuracy over the baseline method (Figure 2b). This result indicates that SM-CNN is more effective when learning from datasets with smaller sizes and may lose its gain in the presence of fully sufficient training data.

## 1.3 Sparsity Measures

This section provides the definition of the sparsity and decorrelation measures:

Figure 2: Validation accuracy of the ResNet experiments on ImageNet with the training data of sizes $50K$ (left) and $10K$ (right).

- Selectivity index [4, 5]:

$$S_S = 1 - \frac{N\bar{r}^2}{\sum_{i=1}^{N} r_i^2} \tag{1}$$

- Kurtosis index [3]:

$$S_K = \frac{1}{N} \sum_{i=1}^{N} \frac{(r_i - \bar{r})^4}{\sigma^4} \tag{2}$$

- Gini index [2]:

$$S_G = 1 - \frac{2}{N^2\bar{r}} \sum_{i=1}^{N} r_{(i)}(N - i + \frac{1}{2}) \tag{3}$$

- Separation angle [4]:

$$\theta(p,q) = cos^{-1}(|p^T q/(\|p\|_2\|q\|_2)|) \tag{4}$$

in which $r$, $p$, and $q$ denote the vector of neural responses, $\bar{r}$ and $\sigma$ are the mean and standard deviation of the neural responses, respectively, and $r_{(i)}$ is the $i$th element of the ordered data $r_{(1)} \leq r_{(2)} \leq ... \leq r_{(N)}$. To calculate the Gini and selectivity indices, we used the amplitude of neural responses, but in the case of the kurtosis index, we appended the negative of each vector to it and then calculated the kurtosis index.

## 1.4 Lifetime and Population Sparsity Distributions

In this section, we report the average of the lifetime sparsity across the population and the trials (Table 1), and the average of the population sparsity across the stimuli and the trials (Table 2). The results show that surround modulation and Batch Normalization significantly increase the lifetime and population sparsity as measured by the different criteria ($p < 0.001$). Figure 3 shows the distributions of the lifetime sparsity and the population sparsity of the first convolutional layer, which were not included in the paper.

## 1.5 Separation Angle Distribution

In the paper, we followed the formulation offered by [4] to calculate separation angles between each pair of response vectors for the baseline$_{\mathrm{BN1}}$, SM-CNN, and baseline networks. However, this criterion is sensitive to the distribution of responses which varies in different models. To compare

(a)                                                      (b)

Figure 3: Distributions of the lifetime sparsity (left) and population sparsity (right) of the first convolutional layer.

Table 1: Average lifetime sparsity scores across 20,000 neurons and six trials.

|  | First convolutional layer | | | Third convolutional layer | | |
|---|---|---|---|---|---|---|
|  | baseline | baseline$_{BN1}$ | SM-CNN | baseline | baseline$_{BN1}$ | SM-CNN |
| kurtosis | $4.8_{\pm 3.9}$ | $18.3_{\pm 39.5}$ | $8.4_{\pm 5.5}$ | $8.0_{\pm 11.5}$ | $19.2_{\pm 36.3}$ | $24.4_{\pm 38.6}$ |
| selectivity | $0.42_{\pm 0.23}$ | $0.75_{\pm 0.09}$ | $0.57_{\pm 0.20}$ | $0.58_{\pm 0.17}$ | $0.66_{\pm 0.14}$ | $0.74_{\pm 0.15}$ |
| Gini | $0.46_{\pm 0.20}$ | $0.74_{\pm 0.07}$ | $0.58_{\pm 0.17}$ | $0.61_{\pm 0.15}$ | $0.63_{\pm 0.13}$ | $0.72_{\pm 0.14}$ |

different models more impartially, here we calculate the separation angle using the Pearson correlation coefficient:

$$\theta(p,q) = cos^{-1}(|\rho(p,q)|) \tag{5}$$

where $\rho(p,q)$ indicates the Pearson correlation coefficient between neural responses $p$ and $q$. The distribution of separation angles for each layer and each model is presented in Figure 4 and Table 3. Surround modulation, as well as Batch Normalization, strongly decorrelate neural responses between pair of neurons ($p < 0.001$), and this property is also preserved in higher layers.

(a)                                                      (b)

Figure 4: The effect of surround modulation and Batch Normalization on the distribution of separation angles between pairs of neurons in the first (left) and third layers (right). Higher separation angles for a population of neurons indicate that the information carried by each neuron is less correlated with that of other neurons.

Table 2: Average population sparsity across 5000 stimuli and six trials.

| | First convolutional layer | | | Third convolutional layer | | |
|---|---|---|---|---|---|---|
| | baseline | baseline$_{BN1}$ | SM-CNN | baseline | baseline$_{BN1}$ | SM-CNN |
| kurtosis | $5.1_{\pm1.5}$ | $22.7_{\pm13.8}$ | $20.6_{\pm8.2}$ | $8.0_{\pm2.8}$ | $15.5_{\pm5.5}$ | $21.0_{\pm8.9}$ |
| selectivity | $0.48_{\pm0.09}$ | $0.76_{\pm0.05}$ | $0.70_{\pm0.07}$ | $0.63_{\pm0.06}$ | $0.70_{\pm0.08}$ | $0.76_{\pm0.04}$ |
| Gini | $0.51_{\pm0.08}$ | $0.74_{\pm0.03}$ | $0.65_{\pm0.07}$ | $0.63_{\pm0.05}$ | $0.65_{\pm0.09}$ | $0.74_{\pm0.05}$ |

Table 3: Average of separation angles across 1M pairs of responses and six trials.

| | First convolutional layer | Third convolutional layer |
|---|---|---|
| baseline | $83.9_{\pm5.3}$ | $83.9_{\pm5.9}$ |
| baseline$_{BN1}$ | $86.9_{\pm3.4}$ | $87.3_{\pm3.0}$ |
| SM-CNN | $87.2_{\pm4.8}$ | $87.5_{\pm4.9}$ |