[Reviews · NeurIPS 2019]

Reviewer 1



The authors report increased accuracy and robustness by adding an extremely simple, linear filtering with a difference-of-Gaussian kernel. While this would be a quite remarkable finding if it turned out to be reliable, I have substantial doubts, which I will outline below: - Instead of using a well-established baseline (e.g. a ResNet), the authors train their own architecture (similar to VGG) on what seems to be their own 100-class variant of ImageNet. Despite these simplifications, their performance is quite poor (a ResNet-18 would achieve ~70% top-1 accuracy on full ImageNet). As a consequence, we don't really know whether the improvement by their surround modulation module is an artefact of a poorly trained model or a real effect. Training a ResNet on ImageNet is trivial these days (scripts are provided in the official PyTorch or Tensorflow repositories), so it's really unclear to me why the authors try to establish their own baseline. In fact, training VGG-like architectures without batch norm is anything from trivial, so it's quite likely that the authors' model underperforms substantially. - The relation between what the authors do and surround suppression in the brain is weak at best. In the authors' implementation, the only neurons contributing are the ones from the same feature map. Such specificity is unlikely to be the case in the brain. In addition, biological surround modulation is purely modulatory, i.e. has no effect if there is no stimulus in the classical receptive field. However, the authors' implementation as a linear filter will elicit a response if only the surround is modulated without any stimulus in the center. - I am unsure about the value of the analysis about sparsity. What are these figures meant to tell us? I think they could as well be left out without the paper losing anything. - Why is the surround modulation module added only after the first layer? - Were the networks used for NORB and occluded MNIST trained from scratch on these datasets or pre-trained on ImageNet and only fine-tuned?

Reviewer 2



This paper incorporates surround modulation neural mechanism to convolution neural network. The authors add local lateral connections (defined in eqn 1 and 2) to the activation maps of convolutional neural networks that mimics surround modulation. To my knowledge, this work is of interest to both the machine learning and neuroscience communities. I find that the results that surround modulation improves CNN performance very interesting. However, I have the following comments: 1- The application of the difference of Gaussians kernel in the computer vision domain is not new. DoG filtering has been proposed before for feature extraction in computer vision (see for example Lowe et al., IEEE 1999 also a wikipedia article here https://en.wikipedia.org/wiki/Difference_of_Gaussians and https://en.wikipedia.org/wiki/Scale-invariant_feature_transform). However, I still believe the results that using this kind of 'engineered' filters gives better performance than baselines is interesting finding. 2- The experiments presentation requires more clarity. In particular, I found it hard to understand the baselines. There are three things that one needs to control for: 1) the number of trainable parameters 2) the depth of the network 3) the structure of the SM kernel. The authors tried to clarify their baselines in paragraph starting line 155. However, I find this description largely unclear. 3- The experiments lack hyperparam tuning. One simple explanation of the results is that the training hyper-parameters were not optimal for the baseline models. 4- For the generalization results, it is unclear whether one could get the same SM-CNN results or better by using standard regularization methods. 5- It would be very interesting to see if the same results would hold for larger networks.

Reviewer 3



Having read the other reviews and the authors' response, I am willing to downgrade my score a tad (due to Reviewer 1's points). But that's still a good score! I didn't quite understand their response to my review; I don't believe I said anything about different size receptive fields. This paper presents an incredibly simple idea that is effective. Given this, I am not sure whether it has been done before (I don’t know of any other papers that do this exact thing, but I am willing to be corrected). Hence I am unsure of the originality. The idea is simple: surround modulation is a pervasive feature of the visual system. A similar surround will reduce the response of a neuron, indicating that nearby neurons sensitive to the same pattern are inhibiting its response. This is implicated in a large number of phenomena in visual neuroscience, which are listed in the paper. There are at least three potential mechanisms for it, one of which is lateral connections with a difference-of-gaussians (DOG), or center-surround shape. Neurons responsive to the same stimulus nearby enhance response, and a little farther away inhibit it. In this paper, the authors choose to implement the lateral inhibition, DOG idea. This is implemented as a convolution of a DOG linear filter on the activation maps of half of the first layer of convolutions (before ReLU, I believe). Other variants are also tested, including applying it to all of the initial filters, applying it to the input image instead of the first layer of convolutions, and applying it to the first maxpooling layer. As a control experiment, another layer is added above the first layer, both with and without the ReLU nonlinearity. Using a subset of ImageNet, they show that this model learns more quickly and achieves higher performance than the control networks. Two of the variants also perform better than an equivalent convnet. Here, the paper would be clearer by associating the three variants with the labels used in Table 1, although it was clear to me. They then test the robustness of this model against different lighting conditions using the NORB dataset. The results are quite convincing, in that the network with the DOG convolution is much more robust to the lighting changes than a network without it, by around 15%. My guess is that one should attribute most of this to the DOGs providing a kind of contrast normalization. They also show that this network is more robust to occlusion in MNIST when compared to a standard convnet. Finally, they show that the network activations are relatively sparse, in both lifetime and population senses, and that the activations are made more independent by this manipulation. The paper is very well-written and clear. Lines 62-77 could be considerably shortened, considering the audience is very familiar with these concepts. I think this paper is significant (modulo my lack of knowledge of it being done before), because it is a simple idea that appears to aid in learning, classification performance, and robustness. I also like the fact that it’s biologically inspired. Line 51: insert "are" between "and" and "unlikely." Line 93: reference 39 is also appropriate here (with 37 and 60). Line 23 of supplementary material: separation is misspelled.

[Author Response · NeurIPS 2019]

We thank the reviewers for their valuable suggestions and critiques and provide clarifications for their comments here.

**First Review:**

• *TECHNICAL ISSUES:* In the ImageNet experiment, we only used 500 samples from each class and did not perform any data augmentation during training or multi-cropping during the test. This was done to provide a standard and balanced dataset and a general CNN architecture, i.e., an impartial setting, to evaluate the exclusive effect of adding SM to the network. To follow the suggestion of the respected reviewer, we repeated the ImageNet experiment with a **standard ResNet-18** (with pre-activation and $L_2$ regularization) and used the same strategy for incorporating the SM kernel. Although ResNet extensively uses batch normalization (BN), which is expected to potentially reduce the gain offered by SM, our SM-ResNet-18 still showed superior performance over the standard ResNet-18 (Fig. a). We repeated this experiment on a smaller dataset obtained by sampling 100 instances from each class. We observed that SM-ResNet-18 offers about $12\%$ gain in relative accuracy over the baseline. This result indicates that SM-CNN is more effective when learning from datasets with smaller sizes and may lose its gain for very large training datasets. In our implementations, SM was added to the 1st layer as such modulation is more common in the early visual cortex. However, the paper also reports on the result of an SM-CNN variant with SM added to a later layer with results beating the standard CNN. Finding the optimal structure for the SM and its most effective placement in CNN can be studied in the future. We trained all of the models from scratch to better analyze the sole effect of SM on the training procedure.

• *BIO-INSPIRED:* As explained in the introduction, surround modulation (SM) occurs when the center and the surround of receptive fields carry similar visual features. Since the units of each feature map in CNNs contain responses to the same features, we implemented SM by incorporating units of each feature map. We were aware that in our linear model, the center might be suppressed even when it is not activated. To account for this, in our first design, the SM unit was implemented as a linear filter followed by a ReLU nonlinearity, which resulted in a more biologically plausible neural activity. However, experiments showed that a linear filter alone offers slightly better accuracy while being simpler.

• *SPARSITY:* One notable aspect of our study is that the SM structure alters neural activities of CNNs and makes them more biologically plausible. The effect of SM on decorrelation and sparsity of neural coding has been widely studied in neuroscience. We have shown that our implementation of SM results in similar outcomes, even though we did not explicitly design it for this purpose. We included the histograms of sparsity and decorrelation to show how our model agrees with those studies (see Gallant et al. 2000, 2002). These diagrams can be summarized in the final version if recommended. Our analysis of sparsity also motivates the question of whether sparsity in neural activity can boost training, another attribute studied in neuroscience (see Yao et al. 2007). As our paper reports, this notion is supported in our study. In other words, the gain in learning speed, especially when using fewer training samples (Fig. a), made our network behave more analogous to the biological systems, even though the method was not designed for such purpose.

**Second Review:** The application of the DoG filter in computer vision is in edge detection. We examined this in our experiment and did not find it helpful (see the $C_1$ variant of SM-CNN in Table 1). We will add a note about this point. In the paper, we provided 2 control models for the baseline and 3 control models for the SM-CNN, to which we will need to add explicit references to Table 1 in the final version. On the issue of control model details, Supp. Materials contains a description about the structure of the networks, from which the effect of each control model on the # of parameters can be easily derived (these effects will be described in the final version). Hyperparameters were roughly tuned for baselines. Our analysis shows that a learning rate of $10^{-4}$ offers a good tradeoff between speed and robustness for all scenarios. Changing the value of the learning rate does not impact the superiority of our model over the baseline.

• *SUGGESTED CONTROLS:* We designed a new control model based on the suggested **random kernel** matching the 1st and 2nd statistics of the SM kernel, and analyzed it on the ImageNet experiment. The final accuracy was $39.5\%$, which is lower than both the baseline ($43.2\%$) and SM-CNN ($40.9\%$). We also repeated the experiment on small NORB with 3 control models by adding standard **regularization methods** to the baseline network. In the 1st model, we added Dropout to 3 layers. In the 2nd model, we added BN to all convolution layers. In the 3rd model, we added $L_2$ regularization to all of the weights. We extended Fig. 4 of the paper by adding these results to it (Fig. b), illustrating that SM-CNN has better generalization in this problem. As mentioned earlier, Fig. a contains the **ResNet** analysis.

**Third Review:** Thank you for the helpful suggestions. Grating patches with different radius sizes or orientations between surround and center have been widely examined in neurophysiological reports. Performing such analysis and evaluating the similarity of tuning curves with those reports can certainly be targeted as a direction for future work.

(a) ResNet experiments on ImageNet.

(b) Regularization experiments on small NORB.

[Meta-Review · NeurIPS 2019]

This paper adds static linear surround modulation to deep convolutional networks. The authors show that this improves the speed and performance of the networks. They also show that it (but also batch normalization) increases sparsity of neural activity. The performance gains seem to be reduced on larger problems which limited projected significance and excitement in the paper. Limiting suppression to only identical features was also seen as limiting. Reviewers did recognize that training on full ImageNet is computationally expensive.